# Family Structure and Family Climate in Relation to Health and Socioeconomic Status for Older Adults: A Longitudinal Moderated Mediation Analysis

**DOI:** 10.3390/ijerph191811840

**Published:** 2022-09-19

**Authors:** Enrique Alonso-Perez, Paul Gellert, Michaela Kreyenfeld, Julie Lorraine O’Sullivan

**Affiliations:** 1Charité–Universitätsmedizin Berlin, Corporate Member of Freie Universität Berlin and Humboldt-Universität zu Berlin, and Berlin Institute of Health, Institute for Medical Sociology and Rehabilitation Science, Charitéplatz 1, 10117 Berlin, Germany; 2Social Policy Groups, Hertie School, Friedrichstrasse 180, 10117 Berlin, Germany

**Keywords:** family climate, family structure, socioeconomic status, health determinants, biopsychosocial, SHARE

## Abstract

Family characteristics are associated with individuals’ health and wellbeing. However, the link between family structure (e.g., operationalized via marital status) and health outcomes is ambiguous, and whether family climate mediates the relationship is unclear. This study uses the Biobehavioral Family Model (BBFM) to investigate the association of older adults’ family structure with later health, the mediating role of family climate and mental health and how these links vary by socioeconomic status (SES). Using data from *n* = 29,457 respondents aged over 50 in Waves 4, 5 and 6 (2011, 2013 and 2015) of the Survey of Health, Retirement and Ageing in Europe (SHARE), the BBFM was applied in a longitudinal mediation analysis of family structure and health, including both indicators of mental and physical health. Structural equation modeling was applied, and a multigroup analysis was performed to test the role of SES in a moderated mediation. Family climate and mental health mediated the relationship between family structure and subsequent physical health. Good levels of family climate were found to be consistently associated with improved mental and physical health. These relationships were significantly moderated by SES, showing that the association of family climate and health was weaker for those in low SES positions. Family climate and mental health should be considered as potential mechanisms linking family structure to later physical health outcomes across time; however, these associations are diminished for those with low SES.

## 1. Introduction

Family constitutes the context of an individual’s health and is a major social determinant of health, meaning that it comprises the main environment where individuals develop, practice and solidify health behaviors [1]. Coupled with this, the study of social determinants of health plays an important role in informing researchers and policymakers to improve public health [2]. As argued by the World Health Organization (WHO) and its former Commission on Social Determinants of Health, “the conditions in which people grow, live, work and age” and “the fundamental drivers of these conditions” are health-shaping experiences that contribute to the overall health and disease of individuals [3]. Many of these conditions, including material circumstances [4] as well as social [5,6], psychological [7], and biological endowments [8] are closely linked to the current family situation [9]. Although a substantial body of literature has highlighted the importance of these factors as determinants of health, the underlying mechanisms of these links remain to be fully clarified. Furthermore, as aging societies increasingly rely on families to take on care responsibilities for those in need [10], we need to understand more about family dynamics and how they can affect individual health.

Throughout the course of life, the vast majority of individuals are part of family configurations [11], either as children, parents, step-parents, partners or other roles. Social relations with significant others such as family or close friends can yield positive and negative health effects, depending on their frequency, quality and intensity [12]. Recent evidence shows that family relationships tend to have a greater impact on health than those with other people [13]. Furthermore, health behavior, which is known as an important predictor of health [2] is acquired at the level of the family of origin [1]. Thereupon, development of habits and social behaviors are closely related to family characteristics and family interpersonal relations [12,14,15]. Nonetheless, despite growing research on how family and health are connected [16,17], both theoretical debates and empirical studies fail to adequately address an essential yet complex social factor that shapes our health and wellbeing: family climate.

Family climate is conceptualized as the emotional valence and intensity of interpersonal relations and cohesion between family members [18,19]. While previous research found family structure, operationalized over marital status [11], to be a determinant of mental and physical health for all family members [20,21,22], little is known about the role of family climate in this regard. Findings from two recent studies indicate that family climate could play a more important role than family structure, by showing that family climate had stronger effects than family structure in predicting health [18,23]. Furthermore, these studies have made a theoretical distinction between the bilateral relationship quality of specific family members (e.g., partner-partner or parent-child relationships) and family climate involving relationships with the extended family as well, thus expanding the concept of family climate beyond dyadic relationship concepts.

Demo and colleagues [24] argue that the most decisive attributes for defining families are long-run committed relationships and the intertwined responsibilities and support associated with them, as legal marital status or biological ties between family members alone are not sufficient to describe the family experience entirely. In line with this, a few studies pointed out that merely having a partner relationship is not exclusively associated with better psychological wellbeing or health compared to having no relationship [25,26]. Furthermore, although marriage is considered to lead in positive health outcomes, increased strain or low-quality marriages can have detrimental effects on health [27,28]. Taken together, these findings underscore the importance of family climate and implicit family practices. Thus, family structure (e.g., operationalized over marital status) alone may be insufficient to understand experience in family, and we need to consider it jointly with family climate to evaluate effects on health.

Regarding the study of family climate and health, the Biobehavioral Family Model (BBFM) was developed as a biopsychosocial approach that integrates interactive social processes and family climate affecting psychological and physical health [29]. It addresses the association between family relations and health by theorizing a mediation relationship between family climate and physical health through a mental health emotional pathway, namely biobehavioral reactivity [19]. Previous research has expanded the BBFM with the inclusion of social support [30], allostatic load [31], health behaviors [32,33] or racial/ethnic discrimination [34,35] as additional mediating mechanisms. However, when reviewing this line of the literature, it becomes apparent that existing studies have failed to measure both family structure and family climate simultaneously to examine their effects on health outcomes. Furthermore, barring some recent longitudinal investigations [36,37,38], most of the BBFM applications have focused on cross-sectional data. Since family interactions constitute a large part of our daily contexts depending on the phase of the life course we are in, and family members experience many and diverse exchanges as they advance over life, a longitudinal approach is necessary to better understand the temporal interrelations of family structure and family climate on later health. By doing so, a proper sequence of events can be established to detect developments or changes in health outcomes. Therefore, an extension of the BBFM by including family structure and analyzing the longitudinal pathways is relevant to discern the repercussion of family on health.

Additionally, socioeconomic status (SES) is known to be a sound cause of stratification of any given predictor-health relationship [39,40]. Moreover, a causal link has also been established between lower SES and altered family structures [41,42,43]. Considering these effects simultaneously, Goldman [44] discusses effects of SES in the links between family structure and differences in health; hence, it is of interest to investigate how the SES gradient shapes family effects on health. Particularly, while intact family structures can provide a protective factor for health outcomes, this relationship may be affected by further demographic factors such as age, sex or SES [45]. Researchers have provided comprehensive evidence of this principle by finding smaller protective health effects of family structure for groups with lower SES [46,47], or of intact marital status providing a stronger health gain for people in high SES positions [48]. Consequently, it is plausible to consider SES as a moderator of the relationship between family climate and health, although it has been rarely studied [49]. As discussed by Booysen and colleagues [50], the existing body of empirical evidence has investigated these associations separately, and there is a lack of studies on the joint pathways between SES, family structure and health. To deal with these complex mechanisms, they propose a causal model where SES may moderate the links between family structure and health.

Overall, previous research suggests that being married is related to better health outcomes [51,52]. However, the mechanism linking family structure to subsequent health is overlooked by simply studying links between marital status and health, while not thoroughly examining the interpersonal relationships. In line with this notion, there is an emerging body of literature providing evidence that family climate is a decisive determinant of health. Studies incorporating both family structure and family climate however are lacking. Since there is an interlaced relationship between family structure and the valence and intensity of interpersonal relations within families, we aimed to use family climate to unravel the relation of family structure with later health outcomes. Moreover, we intend to investigate differences between SES groups by combining the BBFM and the moderated mediation model proposed by Booysen et al. [50].

To address these aims, we developed a prospective study using data from three successive waves (2011, 2013 and 2015) of the Survey of Health, Ageing and Retirement in Europe (SHARE). In line with previous research, we operationalized family structure over marital status [21,53], with family climate being measured with an adapted scale of family connectedness and cohesion [54]. The latter is based on a subjective definition of family membership and considers both the close and extended family (i.e., family members living in the same household and those beyond the nuclear family), which allows the capturing of familial interpersonal relationships for all marital statuses. Additionally, we include validated measures of biobehavioral reactivity (i.e., symptoms of depression and anxiety) and disease activity (i.e., self-rated health and number of chronic conditions). To the best of our knowledge, this is the first study to design a longitudinal moderated mediation model with these specific variables and to expand the BBFM by adding family structure and SES. Specifically, we hypothesized the following:(a)The longitudinal relationship between family structure (i.e., marital status) and later disease activity (i.e., physical health) is mediated by family climate and biobehavioral reactivity (i.e., mental health).(b)Socioeconomic differences alter the impact of family climate on health outcomes, with weaker health gains for those with lower SES.

## 2. Materials and Methods

### 2.1. Data and Sample

This study uses data from *n* = 29,457 respondents who participated in Wave 4 (2011) and were followed through in Waves 5 (2013) and 6 (2015) in SHARE. This is the largest pan-European social science panel study for people aged 50 or more, with in-depth face-to-face interviews using computer-assisted personal interviewing (CAPI) technology that provide multidisciplinary longitudinal data on health, economic and social factors [55]. Respondents were carefully selected in each country by national survey agencies, and refreshment samples are drawn regularly to compensate for attrition and maintain representation of the younger birth cohorts. The SHARE study has been extensively described elsewhere [56]. Respondents from waves 4 to 6 were located in 12 out of 20 countries participating in SHARE: Austria, Germany, Sweden, Spain, Italy, France, Denmark, Switzerland, Belgium, Czech Republic, Slovenia and Estonia. The average individual response rate was 44.8% in wave 4, varying from 33.0% in Czech Republic to 58.4% in Estonia [57]. We opted to analyze these waves since they contain specific questionnaires about respondents’ social networks and their interpersonal relationships with them; hence, we could derive information on family climate. Moreover, our proposed longitudinal mediation model requires three time points, therefore these were the most recent standard waves that had appropriate information regarding exogenous predicting variable, mediator, and outcome. Regarding the dropout rate, wave 4 included 48,127 participants with variables of interest available at baseline, wave 5 included 38,983 participants (81.0% retained from the baseline sample), and wave 6 included 29,457 participants (61.2% retained from the baseline sample), the latest being our final sample. The following methods and results sections will refer to Wave 4, 5 and 6 as Time 1 (T1), Time 2 (T2) and Time 3 (T3) for ease of presentation. The Ethics Council of the Max Planck Society for the Advancement of Science granted SHARE’s Ethics approval.

### 2.2. Measures

#### 2.2.1. Family Structure

To operationalize family structure, we used the marital status and distinguished between married, divorced, widowed and single (never married). The reference category was married. This variable was measured at T1 and thus treated as an exogenous variable. Family transitions that occurred during the panel were disregarded as we focused on how initial family structure can predict later health. We opted for a single predictor variable to facilitate interpretation while controlling for other family characteristics (see Section 2.2.6). Previous research with similar operationalizations already established the important role of marital status on current and future health [20,21,22,53].

#### 2.2.2. Family Climate

To assess family climate, which serves as a mediator, we applied SHARE’s broad definition of family that is based on the subjective judgement of who respondents consider family. Respondents were asked to classify important family members using a standardized list containing both members of the nuclear family (e.g., partner, parent, or child) and the extended family (e.g., cousin, in-laws, ex-partners etc.). We employed variables from the social network module at T1, where individuals were asked questions about their relationship with up to seven important people. We selected respondents who named at least one family member within their network; hence, only individuals with at least one family member in their network were part of our sample. We created a latent construct with items referring to the interpersonal relationship with their families. Latent variables are inferred from a set of conceptually related items, representing the shared variance of these. They include specific assessments of measurement error, a key assumption that allows to work with subjective measures such as unobservable variables while minimizing the potential measurement error from survey data [58]. Therefore, we used five indicators that measure a latent factor of family connectedness (scale Cronbach’s α = 0.886): number of family members, proximity (number of family members living less than 25 km apart), contact (number of family members with at least weekly interaction), emotional closeness (number of family members with whom they feel very or extremely close) and number of different types of family relationships. We performed an Exploratory Factor Analysis (EFA) to evaluate the unidimensional structure of the measured score and the five items loaded on a single extracted component (KMO = 0.818, Bartlett’s test significant with *p* < 0.001), meaning a single latent factor was sensible. Higher scores represented better family climate. This measure has been used before to capture social network connectedness with SHARE data [54,59] and is consistent with the previous literature on family climate or cohesion [18,60].

#### 2.2.3. Biobehavioral Reactivity

We created a latent construct to represent emotion dysregulation by employing two validated indexes on depression (EURO-D) and anxiety (reduced Beck’s Anxiety Inventory, BAI) measured at T2 (Cronbach’s α = 0.764), similar to previous studies using the BBFM [30,61]. In our model, biobehavioral reactivity was conceptualized as a second mediator. The EURO-D is a 12-item scale developed to measure late-life depression across European countries, and it has shown high correlation to other mental health measures. The scale ranges from 0 to 12, with higher scores representing more depressive symptoms. Similarly, the reduced BAI is composed by 5 items, and it has been evaluated as reliable [62].

#### 2.2.4. Disease Activity

Disease activity was measured as a latent construct with two items: self-perceived health and number of chronic conditions at T3 (Cronbach’s α = 0.855). The former was measured using the US version Likert scale, so that higher values would equal worse health (1 = excellent, 5 = poor). While self-reported health tackles difficulties associated with the collection of comprehensive health data, it involves different assessments of the same status given diverse expectations, scales, or experiences [63]. To obtain a comprehensive measure of disease, we complemented it by adding the number of diagnosed chronic conditions. Previous studies applying the biobehavioral family model used similar constructs as dependent variables [30,31].

#### 2.2.5. Socioeconomic Status

We used education and income to operationalize SES [64], which was included in the models as a moderator. Education was measured by ISCED 1997 (since the 2011 version was still not adapted at T1) and coded into three categories: 1 = lower education (pre-primary or primary education), 2 = medium education (lower or upper secondary education) and 3 = higher education (first or second stage of tertiary education). Household income resulted from either a value directly reported by respondents, their approximation with given bracket values, or in absence of these, from the available imputations in SHARE, whose methodology is described elsewhere [65]. It is assumed that at least some sharing of resources takes place within households; hence, we selected household monthly income in line with our study of families [66]. We adjusted the income by current purchasing power parity (PPP) exchange rates, which harmonize all currencies and PPP throughout the years’ inflation. Finally, education and income were combined to create a final joint SES variable to be used in our analyses, which was categorized into three groups (i.e., low, mid and high) as suggested by previous research on SES as a health determinant [67].

#### 2.2.6. Potential Confounders

We included family support (i.e., care received), family responsibilities (i.e., care provided) and presence of children in the family (no children = 0, one or more = 1) as measures of family caregiving at T1. The first two items were measured with the number of family members who provided help and the number of family members who received care by the respondent. In addition, we considered sociodemographic characteristics available at the first measurement point T1 including gender (men = 0, women = 1), age, migration background (none = 0, born in a country different than the interview = 1), living arrangement (living alone = 0, cohabiting with one or more persons = 1) as covariates for the two models. Age was included as a control for all latent variables. Additionally, in models where SES was not used as a moderator, it was included as a covariate.

### 2.3. Statistical Analysis

To determine the associations between family structure (four dummy coded variables), family climate (continuous) and later biobehavioral reactivity (continuous) and disease activity (continuous), we performed a series of longitudinal mediation analyses using Structural Equation Modelling (SEM) [68] in AMOS^®^25.0 (IBM SPSS, Chicago, IL, USA) and Stata^®^14.0 (Statacorp, College Station, TX, USA). First, we carried out descriptive analyses to illustrate the characteristics of our study sample and variables of interest, including frequencies, means and standard deviations. Subsequently, we conducted Confirmatory Factor Analyses (CFA), where the goal was to assess if the latent constructs were appropriate measures and if the SEM model fitted the data. To determine the goodness of fit, we used the recommended model fit indices Root Mean Square Error of Approximation (RMSEA), Comparative Fit Index (CFI) and Standardized Root Mean Square Residual (SRMR). Threshold values of 0.08 or less for RMSEA, 0.9 or more for CFI and 0.06 or less for SRMR indicate good fit [69]. Subsequently, we ran the SEMs, which included the path analyses based in our hypothesized relationships. We applied the available longitudinal weights in SHARE to procure cross-national equivalence of measurement. These weights were calibrated with total national populations by sex, age and attrition in successive waves, to compensate for unbalanced selection probabilities of different samples [65]. For our SEM analysis, we employed two specifications: to address the first hypothesized association, model (1) included direct paths from family structure to family climate (T1), biobehavioral reactivity (T2) and disease activity (T3), while adding SES as a covariate; and to address the second hypothesis, model (2) was a multigroup SEM analysis to investigate differences in path coefficients between three groups (low SES vs. mid SES vs. high SES), where only direct paths from family structure to family climate were allowed. We were interested in direct effects (i.e., pathway between two variables), indirect effects (i.e., indirect pathway between two variables through a third mediation variable) and total effects (i.e., the sum of direct and indirect effects) as derived from our extended BBFM assumptions. Both hypothesized models are depicted in Figure 1.

We applied longitudinal mediation models due to our hypothesized temporal relationship;, hence, we residualized disease activity at T3 on T1, and biobehavioral reactivity at T2 on T1 to ensure that predictors would predict changes in the mediators, and mediators would predict changes in the outcomes [70]. Regarding the multigroup SEM, we tested the differences in coefficients between groups by analyzing measurement invariance [71]. Path coefficients from the links among latent variables were constrained to be equal, and then model fit compared to the initial unconstrained model. A decrease in model fit is evidence of significantly different coefficients for each group. Additionally, we ran omnibus join tests, including score tests and Wald tests to test if constraint release of certain paths would significantly improve model fit.

Following the missing values analysis, missing data represented 5.2% of total observations. Little’s test of missing completely at random (MCAR), χ^2^ (2716) = 24,174.83, *p* < 0.001, revealed that data were not missing completely at random. We then compared frequencies throughout waves, which demonstrated that more missing data occurred at T2 and T3 than at T1. Considering this pattern and assuming that data are missing at random (MAR), we used the full information maximum likelihood (FIML) estimator, which allows us to counteract missing values by maximizing statistical power while diminishing potential bias [72].

## 3. Results

### 3.1. Sample Characteristics

Table 1 depicts an overview of the sample characteristics at T1 and descriptive statistics of the variables used in our study at T1, T2 and T3. The average age of the selected respondents was 65.40 (SD = 10.4) years old with only 107 respondents aged over 90 years (0.2% of the sample). Women represented 56.73% of the sample. Regarding family structure, the majority were married (70.84%), while widowed was the second most prominent category (14.78%), followed by divorced (8.75%) and never married (5.63%). In terms of family climate, 7204 (13.03%) individuals reported family members beyond the nuclear family, which included grandchildren, as being part of their close social network. Furthermore, respondents reported an average of 1.77 (SD = 0.96) family members with whom they felt very or extremely close. Appendix A present the correlations between key variables.

### 3.2. Baseline Model

The CFA analyzed the measurement model by examining the latent constructs of family climate at T1, biobehavioral reactivity and T2 and disease activity at T3. This model provided good fit to the data: RMSEA (95% CI) = 0.045 (0.044; 0.047), CFI = 0.924, SRMR = 0.037. Similarly, individual manifest variables had factor loadings of >0.40 into the latent variables, meaning that all of them fit together appropriately on the latent construct.

### 3.3. Structural Equation Modelling

Figure 2 presents the results from the first serial mediation analysis with SEM, showing the direct significant paths between variables. This information is complemented in Table 2, which depicts direct, indirect and total effects of all variables on family climate, biobehavioral reactivity and disease activity. With *n* = 29,457 observations for this model, goodness of fit was adequate: RMSEA (95% CI) = 0.051 (0.049; 0.052), CFI = 0.914, SRMR = 0.035. We found that none of the family structure variables at T1 had significant direct or total effects on neither behavioral reactivity at T2 nor disease activity at T3. However, widows, divorcees and singles showed significantly decreased family climate compared to married respondents at T1, with the single group having the biggest reduction (−0.157, *p* < 0.001). Additionally, being widowed, divorced or single at T1 indirectly increased biobehavioral reactivity at T2 through family climate, while indirect significant effects were found for the comparison between being widowed or divorced at T1 and disease activity at T3. These results, given the significant correlation between family structure and later disease activity, support our first hypothesis as family structure affected disease activity across time only through the mediation of family climate and biobehavioral reactivity.

Moreover, direct effects show how good family climate at T1 significantly lowered biobehavioral reactivity at T2 (−0.095, *p* < 0.001) and disease activity at T3, the latter with both significant direct and indirect effects (total effects of −0.041, *p* < 0.001). Aligned with this, biobehavioral reactivity at T2 significantly increased disease activity at T3 (0.102, *p* < 0.001). Therefore, better family climate improves subsequent health outcomes while controlling for all mentioned confounders including SES.

The multigroup SEM results provided in Table 3 exhibit differences across SES groups (see also Appendix A). With *n* = 29,457 individuals included in this model, goodness of fit was adequate: RMSEA (95% CI) = 0.055 (0.054; 0.056), CFI = 0.914, SRMR = 0.039. While all family structure types showed significantly lower family climate at T1 compared to those who were married, higher SES levels lessened this effect. Regarding direct effects on both biobehavioral reactivity at T2 and disease activity at T3, family climate had a protective effect that gradually dwindled for those with lower available socioeconomic resources. Furthermore, the model with constrained parameters for all groups provided worse fit measures: RMSEA (95% CI) = 0.057 (0.056; 0.059), CFI = 0.829, SRMR = 0.052, showing that differences in coefficients between SES groups were significant. In addition, joint tests for each parameter class (Wald and score tests) provided significant differences (*p* < 0.001), thus rejecting invariance across groups. Consequently, respondents with worse SES displayed significantly less health gains from higher family climate compared to those with mid and low SES; hence, the results provide evidence to support our second hypothesis.

## 4. Discussion

The current study applied the BBFM to investigate the longitudinal associations between family structure, family climate and health, the latter being characterized by biobehavioral reactivity (i.e., mental health) and disease activity (i.e., physical health). We considered the mediating role of family climate and biobehavioral reactivity, while adding SES as a potential moderator in a second step. In line with our initial hypothesis, we found evidence of a significant mediation effect of family climate and biobehavioral reactivity in the path between family structure and disease activity across the years. Furthermore, a positive family climate was linked to better mental and physical health in later years. Finally, our results provide evidence that SES moderates the relation between family climate and health, meaning that the beneficial impact of family climate on health is weaker in those individuals with lower SES.

Our findings regarding the important association between family climate and health are aligned with previous studies [18,73], and embedded within the theoretical framework of family as a major social determinant of health. Moreover, the role of intact family structures or better family climate has been documented as associated with better health outcomes [20,22,33]. Our finding that being in an intact relationship was associated with better health compared to other family structures such as being divorced or widowed, supports this research. However, we also found these associations to be less direct and weaker than in previous studies. Precisely, our study reveals that protective health effects of family structures are in fact mediated by family climate. This draws on the conceptual framework that emotional protection provided by marriage and family ties is conditional to the family climate and potential psychosocial support, coupled with the monitoring of health behaviors [16,74]. Previous studies have fallen short of taking a holistic measurement of family characteristics, since they either included only single-item measures of family structure or put all the emphasis on family climate. The present work builds on previous research and complements the theoretical approach by incorporating both family structure and family climate as joint determinants of later health.

### 4.1. Contextualization of Findings

When stressful or unpleasant situations arise, which may well be disruptions in the family climate or those resulting from the myriad processes families experience in daily life, individuals regulate their emotions through biobehavioral reactivity. The family can be either the cause of distress or the solution in this setting, as family climate can influence the impact and direction of such reactivity. Drawing on the BBFM and the theoretical framework formulated by Wood [29], we found empirical evidence that applying a biopsychosocial approach is reasonable, as psychophysiological stress reactions—i.e., biobehavioral reactivity conceptualized by anxiety and depression levels—play a mediating role in conducing the longitudinal effect of family climate on physical health. While existing research applied the BBFM to find similar results, to our knowledge no previous studies have included measures of family structure to expand the model. Our findings confirm that family structure is an important precursor for family climate, while the latter has a central mediating role with regards to health effects across time. Aligned with empirical evidence, individuals who were widowed, divorced or never married showed worse levels of family climate compared to those in a relationship [17].

While there exists a vast amount of literature regarding associations between SES and health, and SES and family structure, little is known about the interrelation between these three constructs. As discussed by Booysen and colleagues [50], empirical work tends to include separate analyses of these variables, therefore not exploring the potential interactions between family structure, SES and health. These authors proposed a causal model that we adopted in our study, since it fitted the theoretical framework of the BBFM and allowed the inclusion of SES as moderator instead of a covariate. Our results do not only provide evidence of the suitability of this conceptual model, but also go further by exploring longitudinal associations mediated by family climate. Particularly, our findings suggest that higher SES has a buffering effect by mitigating the decline in family climate for altered family structures, which is consistent with previous research [44,75]. Aligned with this, the positive association between family climate and health is gradually weakened with lower levels of SES, therefore this group does not benefit much from a better family climate. These results could be explained on the grounds that individuals with lower levels of education and income may have less available resources and abilities to overcome the negative consequences of altered family structures or low family climate [76]. Since other factors drive their relationship between family structure and health, including access to health care or health behaviors related to social norms of peers, inequalities in health remain despite the presence of a good family climate [4].

### 4.2. Strengths and Limitations of This Study

The strengths of the present study consist of a large sample size based on representative survey data, the use of validated scales for predictors and moderators, a longitudinal design that allows to examine the temporal relationship between variables, and the use of latent variables that include explicit assessments of measurement error. Nonetheless, findings should be interpreted in light of some limitations. First, our operationalization of family structure is comparable to the shortcomings seen in other studies, since it is limited to marital status. This is a simplified measurement that is tied to the sample characteristics and does not take into account other factors that may determine family structure, such as previous marriages, marital history or cohabitation. However, we chose to retain marital status as an indicator of family structure to facilitate a straightforward interpretation, and added the presence of children and living arrangements as covariates. Second, the present study of family climate does not include a life course perspective; hence, the dynamics of previous experiences and their effect on current family climate are not fully captured. Moreover, further diversity dimensions such as lesbian, gay, bisexual, transgender/transsexual plus (LGBT+) status or more comprehensive measurements of migration background could not be included due their absence in SHARE. Third, SHARE was not specifically designed for measuring psychological and emotional features of personal interrelations; thus, the construct of family climate could not be derived from a specific validated scale. However, we applied a measure previously used to assess social connectedness with SHARE data. Fourth, the variables we employed to measure physical health are prone to the subjective bias of the respondent or medical professionals. Ideally, the same model should be applied with more detailed health measurements, such as biomarker data, to gain more accurate insights on the health effects. Fifth, although we introduced sex as a confounding variable into our models, we assumed the same model for women and men. Although stratifying by sex could illustrate differences on how family climate affects health, incorporating more group-specific analyses would entail a more complex model complicating the interpretation. Lastly, the present study accounted for dropout by only including respondents who were followed through waves 4 to 6, as well as by using the FIML estimator. Although multiple studies using SHARE longitudinal data followed the same sampling technique, potential selection biases cannot be fully ruled out.

### 4.3. Perspectives and Implications

Our findings have implications for public health and social policy as we explore important mechanisms that could prevent chronic disease as well as promote physical and mental health. It is critical to strengthen and improve family climate through public health interventions. On the other hand, biobehavioral reactivity is found to be a pivotal factor in connecting family structure and physical health; therefore, health initiatives should enhance mental health and promote stress-coping strategies, especially in families with lower resource availability or at risk of non-desired family structure alterations. However, our study also reveals that there is a gap in research regarding the integration SES interdependencies into the context of family and of health. Although we contribute by applying a conceptual model that yields valuable connotations, there remains much to be explored about the theoretical basis of the linkages between family structure, family climate, SES and health. A stronger consolidation of these components would be pertinent, coupled with the development of theoretical models incorporating the BBFM and particular mechanisms connecting health behavior processes and further SES aspects. This study focused exclusively on individuals aged 50 or more, but different mechanisms could apply for younger people. Including younger cohorts might bring a different outlook and help on generalizing our results. It is of special interest to apply a life course perspective, as this could provide more detailed insights regarding the temporal dynamics and mechanisms of family characteristics affecting physical and mental health [77]. Our findings show weaker effects for low SES groups and reveal persisting social inequalities, which imply the need for family and health policies to focus on decreasing the inequality gap from diverse angles beyond family climate and resource availability. Accordingly, an interdisciplinary approach should be adopted to investigate further mechanisms linking family diversity and health inequalities across social positions. Alongside this, multidisciplinary collaborations would be necessary to bridge branches of knowledge and acquire an accurate understanding of these interdependencies.

## 5. Conclusions

This study indicated that family climate is a crucial factor in the process of explaining associations between family structure and health through life. While the inclusion of family climate is gradually expanding in empirical research, most studies have failed to successfully combine it with family structure to explain subsequent effects on health. We derived a biopsychosocial model that provides evidence for the importance of mental health in linking between family climate and physical health. Furthermore, our findings shed light on the socioeconomic differences between families since we found a systematic advantage of high SES profiles regarding health gains through family climate. Given that family climate is associated with clear health benefits and that those are contingent on available economic and cultural resources, public health may benefit from measures aimed at improving family climate, although further actions would be required to address the specific needs of low-SES families and decrease the inequality gap.

## Figures and Tables

**Figure 1 ijerph-19-11840-f001:**
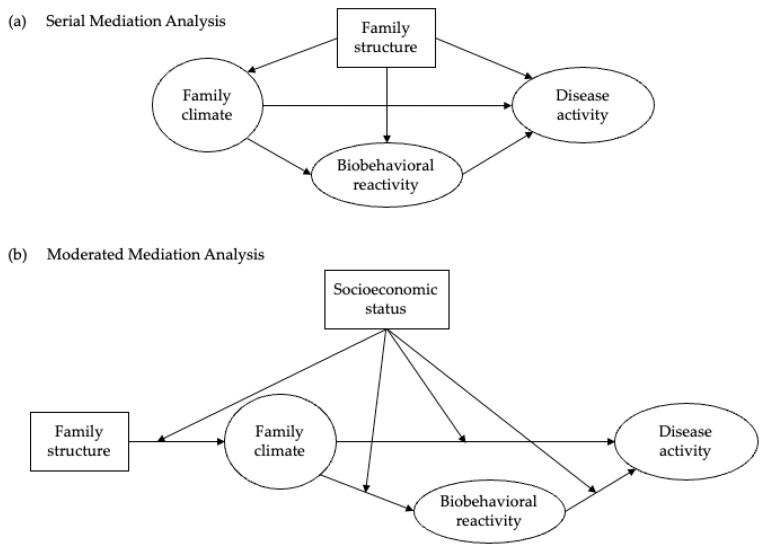
Hypothesized paths for the biobehavioral family model, including (**a**) family structure in a serial mediation analysis and (**b**) socioeconomic status in a moderated mediation analysis.

**Figure 2 ijerph-19-11840-f002:**
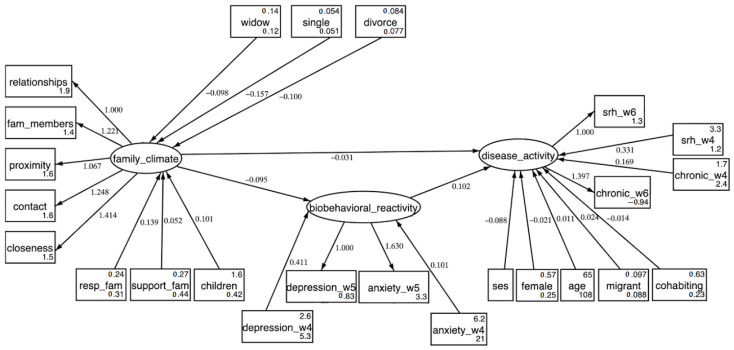
Structural equation modelling (SEM) showing the serial mediation analysis (*n* = 29,457). Rectangles represent manifest variables and ovals represent latent variables. Path coefficients are standardized and only those significant at the *p* < 0.05 level are shown. Covariances among all manifest variables are not shown. All the latent constructs and family structure variables are corrected with age.

**Table 1 ijerph-19-11840-t001:** Sociodemographic characteristics of the sample, Wave 4, 2011 (T1), *n* = 29,457.

	Variable	Category	Values	N (%)
Time 1	Age (mean, ±SD)		50–101	65.44 (10.38)
	Sex	Female		16,711 (56.73)
		Male		12,746 (43.27)
	Migration background	Migrant		2866 (9.73)
		Non-migrant		26,591 (90.27)
	Education	Low		12,275 (41.67)
		Mid		11,329 (38.46)
		High		5856 (19.88)
	Household income	Low tertile		9830 (33.37)
		Mid tertile		9786 (33.22)
		High tertile		9842 (33.41)
	Socioeconomic status (SES)	Low SES		11,061 (37.55)
		Medium SES		10,849 (36.83)
		High SES		7547 (25.62)
	Family structure	Married		20,867 (70.84)
		Widowed		4354 (14.78)
		Divorced		2577 (8.75)
		Single/never married		1658 (5.63)
	Family climate (mean, ±SD)	Diversity of relations	0–4	2.09 (1.12)
		Proximity	0–4	1.75 (0.89)
		Contact	0–4	1.83 (0.88)
		Emotional closeness	0–4	1.77 (0.96)
		Number of family members	0–4	1.65 (0.90)
	Family responsibilities (mean, ±SD)			0.24 (0.56)
	Family support (mean, ±SD)			0.27 (0.66)
	Children	One or more		26,656 (90.49)
		None		2801 (9.51)
	Living arrangement	Cohabiting with someone		18,531 (62.91)
		Living alone		10,926 (37.09)
	Biobehavioral reactivity (mean, ±SD)	EURO-D	0–12	2.59 (2.30)
		BAI-5	0–20	6.21 (4.52)
	Disease activity (mean, ±SD)	Self-rated health	1–5	3.25 (1.08)
		Number chronic diseases	0–11	1.74 (1.55)
Time 2	Biobehavioral reactivity (mean, ±SD)	EURO-D	0–12	2.43 (2.24)
		BAI-5	0–20	5.91 (4.60)
Time 3	Disease activity (mean, ±SD)	Self-rated health	1–5	3.30 (1.05)
		Number chronic diseases	0–11	1.91 (1.64)

SD—Standard Deviation; EURO-D—European Union Depression scale; BAI-5—Beck’s Anxiety Inventory reduced scale.

**Table 2 ijerph-19-11840-t002:** Standardized beta-coefficients from the structural equation modelling (SEM) showing the serial mediation analysis (*n* = 29,457).

Variable	Direct Effect	Indirect Effect	Total Effect
**Family climate**			
Family structure			
Married	Reference	Reference	Reference
Widow	−0.098 *** (0.008)		−0.098 *** (0.008)
Divorced	−0.011 *** (0.013)		−0.011 *** (0.013)
Single/never married	−0.157 *** (−0.010)		−0.157 *** (−0.010)
Support	0.052 *** (0.004)		0.052 *** (0.004)
Responsibilities	0.139 *** (0.005)		0.139 *** (0.005)
Presence of children	0.101 *** (−0.004)		0.101 *** (−0.004)
**Biobehavioral reactivity**			
Family climate	−0.095 *** (0.017)		−0.095 *** (0.017)
Family structure			
Married	Reference	Reference	Reference
Widowed	0.213 (0.017)	0.009 *** (0.002)	0.222 (0.027)
Divorced/separated	−0.006 (0.016)	0.010 *** (0.003)	0.004 (0.032)
Single/never married	−0.056 (0.040)	0.015 *** (0.002)	−0.041 (0.040)
Presence of children		−0.010 *** (0.002)	−0.010 *** (0.002)
Family responsibilities		−0.013 *** (0.002)	−0.013 *** (0.002)
Family support		−0.005 *** (0.001)	−0.005 *** (0.001)
Depression T1	0.411 *** (0.005)		0.411 *** (0.005)
Anxiety T1	0.101 *** (0.002)		0.101 *** (0.002)
**Disease activity**			
Family climate	−0.031 *** (0.007)	−0.010 *** (0.002)	−0.041 *** (0.007)
Biobehavioral reactivity	0.102 *** (0.003)		0.102 *** (0.003)
Family structure			
Married	Reference	Reference	Reference
Widowed	−0.023 (0.013)	0.026 *** (0.003)	0.003 (0.013)
Divorced/separated	0.016 (0.014)	0.003 ** (0.003)	0.019 (0.014)
Single/never married	0.010 (0.017)	0.001 (0.004)	0.011 (0.017)
Support		−0.002 *** (0.001)	−0.002 *** (0.001)
Responsibilities		−0.006 *** (0.001)	−0.006 *** (0.001)
Presence of children		−0.004 *** (0.001)	−0.004 *** (0.001)
Female (Ref: Male)	−0.021 ** (0.001)		−0.021 ** (0.008)
Age	0.011 *** (0.001)		0.011 *** (0.001)
Socioeconomic status	−0.088 *** (0.005)		−0.088 *** (0.005)
Migrant (Ref: Non-migrant)	0.024 * (0.013)		0.024 * (0.013)
Cohabiting (Ref: Living alone)	−0.014 * (0.009)		−0.014 * (0.009)
Depression T1		0.042 *** (0.001)	0.042 *** (0.001)
Anxiety T1		0.010 *** (0.001)	0.010 *** (0.001)
Self-rated health T1	0.331 *** (0.003)		0.169 *** (0.003)
Chronic diseases T1	0.169 *** (0.005)		0.331 *** (0.005)

Note: Standard errors between brackets. The model controls for country and age for all the key variables. * *p* < 0.05; ** *p* < 0.01; *** *p* < 0.001.

**Table 3 ijerph-19-11840-t003:** Standardized beta-coefficients from the multigroup structural equation model analyzing the moderated mediation by group of SES level (*n* = 29,457).

Variable	Low SES	Mid SES	High SES
**Family climate**			
Family structure			
Married	Reference	Reference	Reference
Widow	−0.114 *** (0.011)	−0.071 *** (0.014)	−0.048 * (0.012)
Divorced	−0.176 *** (0.017)	−0.086 *** (0.014)	−0.050 * (0.012)
Single/never married	−0.222 *** (0.020)	−0.163 *** (0.019)	−0.047 * (0.021)
Support	0.073 *** (0.006)	0.039 *** (0.006)	0.052 *** (0.010)
Responsibilities	0.125 *** (0.008)	0.141 *** (0.007)	0.138 *** (0.011)
Presence of children	−0.091 *** (0.005)	0.103 *** (0.005)	0.135 *** (0.005)
**Biobehavioral reactivity**			
Family climate	−0.055 ** (0.028)	−0.077 *** (0.025)	−0.119 *** (0.029)
Biobehavioral reactivity	0.416 *** (0.008)	0.393 *** (0.007)	0.406 *** (0.009)
Female (Ref: Male)	0.122 *** (0.003)	0.094 *** (0.003)	0.071 *** (0.004)
**Disease activity**			
Family climate	−0.014 ** (0.010)	−0.038 *** (0.010)	−0.057 *** (0.014)
Biobehavioral reactivity	0.103 *** (0.004)	0.104 *** (0.004)	0.101 *** (0.007)
Female (Ref: Male)	0.004 ** (0.012)	−0.036 ** (0.011)	−0.058 *** (0.016)
Age	0.013 *** (0.000)	0.011 *** (0.000)	0.010 *** (0.001)
Migrant (Ref: Non-migrant)	0.072 * (0.023)	0.033 *** (0.019)	0.029 * (0.023)
Cohabiting (Ref: Living alone)	0.037 *** (0.012)	−0.003 * (0.011)	−0.036 * (0.016)
Self-rated health T1	0.316 *** (0.007)	0.340 *** (0.007)	0.341 *** (0.009)
Chronic diseases T1	0.152 *** (0.004)	0.179 *** (0.004)	0.198 *** (0.006)

Note: Standard errors between brackets. The model controls for country and age for all the key variables. * *p* < 0.05; ** *p* < 0.01; *** *p* < 0.001.

## Data Availability

SHARE data are publicly available (www.share-project.org, accessed on 5 July 2022).

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
