# Peer review of "Family Structure and Family Climate in Relation to Health and Socioeconomic Status for Older Adults: A Longitudinal Moderated Mediation Analysis"

_ijerph, 2022, doi:10.3390/ijerph191811840_

Round 1
Reviewer 1 Report
The manuscript used 29,457 respondents from the Survey of Health, Retirement and Ageing in Europe (SHARE) to examine whether family climate and mental health mediated the relationship between family structure and physical health. The results indicate that family climate was associated with better mental and physical health and that these relationships were moderated by SES. The topic addressed in this manuscript is potentially important and may contribute to this line of research. However, I have concerns in the manuscript as listed below, that I hope can assist the manuscript.
1. Sample: The sample population was for people aged 50 or more, this need to be reflected in manuscript title and abstract,
2. Family climate measurement: Line 186-189, stated latent variable is free of measurement error, I do not think the statement is correct. In addition, the family climate was measured by items from family connectedness. As you pointed out in the limit section, you did not capture dynamic of family climate in life course perspective, thus, why not call this variable as family connectedness rather than family climate? Have literature supported the measurement of family climate as solely family connectedness? Likewise, for the biobehavioral reactivity measurement, it was indeed a depression measure. Why not use depression directly? The biobehavioral reactivity could be meant other things than depression.
3. Missing data: Line 289-290, stated that “Little's test of missing completely at random (MCAR), χ2 (2716) = 24174.83, p < 0.001, revealed that missing data were due to survey attrition instead of item non-response.” The significant result suggested that the missing was not at random. Moreover, what was the sample size for your SEM analysis? Need to present the size in the Figure and table.
4. Age of sample: ranged from 50 to 111? How many respondents aged over 100 or 90? The family structure such as widowed has quite different meaning at age 50 or 100. Also, how the data were collected for age group older than 90?
5. SEM results: It is unclear whether “family structure affected disease activity through the mediation of family climate and biobehavioral reactivity” (line 327-328), as there was no correlation analyses among variables presented. Was the correlation between family structure and disease activity significant? If not, then you may not have mediation model. Also, the sizes of indirect effects were small.
Reviewer 2 Report
This study discusses the relationships between family structure, family climate, socioeconomic status (SES), and health outcomes. The authors investigate the relationships with the Biobehavioral Family Model (BBFM) using the data from respondents in Waves 4,5, and 6 of the Survey of Health, Retirement and Ageing in Europe (SHARE). Structural equation modeling is used in the analysis, and SES is treated as a moderator. The analysis shows that family climate and mental health mediated the relationship between family structure and physical health, and SES moderated the relationship.
The study is potentially impactful because it is one of the first studies that investigate the family climate as a mediator between family structure and health outcomes. The authors provided sufficient background information in the Introduction Section. It was not challenging to understand the Materials and Methods Section, and sufficient details were provided in the Results Section.
The authors might need to introduce more details about the SHARE studies. How were the respondents selected? In which regions were the respondents located? How were the data collected? What was the response rate?
This study involves a longitudinal analysis, using the waves 4 to 6 data. Dropout is likely to happen during such studies. The authors might need to report the dropout rate and clarify how the dropout samples are used in the analysis. Note that the dropout is not random in this analysis because the respondents with the worst health outcomes are likely to have a higher mortality rate. Therefore, the authors might also want to discuss the potential bias caused by the dropout.
The marriage status might change during the longitudinal study. It is not clear to me how the authors dealt with such situations.
The marriage status is dependent on age (older people are more likely to be a widow). In Figure 2, it looks like the family climate is not corrected with age. I am not sure whether this is proper.
The analysis is based on the data collected from respondents aged 50 or more. The authors might want to discuss whether the results can be generalized to younger people.
Round 2
Reviewer 1 Report
Excellent job on the revision.